# Generation of Lamprey Monoclonal Antibodies (Lampribodies) Using the Phage Display System

**DOI:** 10.3390/biom9120868

**Published:** 2019-12-12

**Authors:** Khan M. A. Hassan, John D. Hansen, Brantley R. Herrin, Chris T. Amemiya

**Affiliations:** 1Department of Molecular and Cell Biology, University of California-Merced, Merced, CA 95343, USA; 2U.S. Geological Survey, Western Fisheries Research Center, Seattle, WA 98115, USA; jhansen@usgs.gov; 3Department of Pathology and Laboratory Medicine, Emory University, Atlanta, GA 30322, USA; Brantley.herrin@gmail.com

**Keywords:** variable lymphocyte receptors, phage display system, library screening, lampribody

## Abstract

The variable lymphocyte receptors (VLRs) consist of leucine rich repeats (LRRs) and comprise the humoral antibodies produced by lampreys and hagfishes. The diversity of the molecules is generated by stepwise genomic rearrangements of LRR cassettes dispersed throughout the VLRB locus. Previously, target-specific monovalent VLRB antibodies were isolated from sea lamprey larvae after immunization with model antigens. Further, the cloned VLR cDNAs from activated lamprey leukocytes were transfected into human cell lines or yeast to select best binders. Here, we expand on the overall utility of the VLRB technology by introducing it into a filamentous phage display system. We first tested the efficacy of isolating phage into which known VLRB molecules were cloned after a series of dilutions. These experiments showed that targeted VLRB clones could easily be recovered even after extensive dilutions (1 to 10^9^). We further utilized the system to isolate target-specific “lampribodies” from phage display libraries from immunized animals and observed an amplification of binders with relative high affinities by competitive binding. The lampribodies can be individually purified and ostensibly utilized for applications for which conventional monoclonal antibodies are employed.

## 1. Introduction

Agnathans diverged from a common vertebrate lineage in the Cambrian period some 500 million years ago [1], with lampreys and hagfish being the only surviving representatives of this group. A unique adaptive immune response mediated by single solenoid-shaped molecules synthesized from highly polymorphic leucine rich repeats (LRR) has been described in lamprey [2] and hagfish [3]. These molecules are expressed in cells resembling B-lymphocytes and are called variable lymphocyte receptor B (VLRBs). They are assembled on an incomplete genomic template (VLRB locus) by stepwise recruitment of the LRR cassettes spread across the large locus [2] to form a mature VLRB (mVLRB), most likely by a gene conversion-like (copy choice) mechanism [4] using very short sequence homology as described for transposon excision in *Escherichia coli* [5]. The system has the theoretical potential of generating a repertoire >10^14^ different antigen receptors, which is comparable to what is seen in the immunoglobulin (Ig)-based adaptive immune system in the jawed vertebrates [3,6]. VLRB monoclonal antibodies, hitherto referred to as “lampribodies”, have been produced using soluble protein, particulate antigens, and whole cells as immunogens, e.g., BclA protein of *Bacillus anthracis* exosporium [7], H-trisaccharide of human O-type erythrocyte [8], hen egg lysozyme [9], avian influenza virus hemagglutinin [10,11], CD38 plasma cells [11], plant pathogen protein HopM1 [12], and murine brain extracellular matrix (ECM) [13]. It is of interest to note that the VLRB system has shown the capability of producing high affinity antibodies against glycotopes, much more so than the immunoglobulin-based system [14]. VLRBs have also made their way into therapeutic and diagnostic applications, including monitoring of chronic lymphocytic leukemia [15], blocking the IL-6/STAT3 signaling pathway in non-small cell lung cancer cells [16], delivery of protein–drug conjugates for tumor regression [17], visualizing protein–protein interaction in live cells [18], suppression of neovascularization in age-related macular degeneration by blocking the VEGF signaling pathway [19], neutralizing anaphylatoxin C5a to combat inflammatory disease [20], and applications in gene therapy with chimeric antigen receptors targeting leukemic B cells and effector T cells [21].

Isolation of the antigen specific VLRB has been accomplished in various ways. These include cloning the PCR-amplified cDNA from leucocytes into an expression vector and transfecting and screening human cell lines [2,15]. VLRB amplicons have also been expressed on the surface of yeast to develop a yeast surface display (YSD) platform for isolating target specific VLRB molecules and for in vitro affinity maturation [9]. Insect cells have been used with the baculoviral expression system in order to secrete VLRB protein [22], and HEK-293T human cells have also been used for this purpose [15]. In these methods, random clones from the respective libraries were individually tested for their affinity to the specific antigen and the best binding clones were isolated for further analysis. Since the first reporting of the phage display system in 1987 [23], it has been used for many proteins including genetically engineered VLRB molecules for screening and selection of binders with relative high affinity [24]. However, the use of the system for generating libraries from immunized lampreys and for clone selection has never been reported.

In this study, our first objective was to evaluate the robustness of the phage display system in VLRB expression and target specific binding, and the second objective was to develop a VLRB screening method based on phage display. First, we used a semi-quantitative means (subtraction experiment) in order to examine its sensitivity by isolating known VLRB molecules embedded within very large libraries. Secondly, the system was challenged (by competitive binding) to select target specific VLRB from libraries of immunized animals. Our results collectively demonstrate that the phage display system is suitable for isolating the highest affinity VLRB molecules from libraries, resulting in clonal expansion of such molecules. The lampribodies so generated should be usable for many applications in biology and are highly amenable to an open source model of communal distribution.

## 2. Results

### 2.1. Testing the Driver VLRB Molecules in M13 Phage Display System

In order to examine target specific binding of the VLRB molecules expressed in a phage display system, we used the previously published variable regions of anti-lysozyme VLRB molecules, VLRB.HEL.1, VLRB.HEL21, and VLRB.HEL2D [9]. Full-length anti-lysozyme VLRB molecules were synthesized from the respective variable regions by adding the conserved 5′ and 3′ regions of VLRB. The synthesized molecules were amplified and subsequently cloned into the phagemid vector, pADL-10b, yielding a fusion protein with the pIII coat protein of the phage as described schematically in Figure 1A,D using primers in Appendix A.

M13 phages were prepared using Hyperphage (M13K07 ΔpIII) [25] in TG1 *E. coli* cells. To test the relative binding affinity, approximately 1.0 × 10^9^ plaque forming units (PFU) of each phage were used in ELISA detection of two-fold serial dilutions on streptavidin microtiter plates coated with biotinylated lysozyme. The results (Figure 2A) clearly showed binding of phages expressing VLRB-pIII recombinant protein to hen egg lysozyme. The binding affinity of VLRB.HEL21 phage was higher than that of VLRB.HEL1 and VLRB.HEL2D phages; approximately 0.97 million PFU of recombinant phages or more showed appreciable optical density (OD) readings (>0.1 OD) for each of these phages. No appreciable binding of any of the phages was seen in wells without bait and the OD values were comparable to those obtained with blocking buffer with and without bait. It demonstrated that the binding affinity of the phages to plastic was equally negligible as that of the blocking buffer. Based on this result, these VLRB clones were considered suitable for use as drivers in the assessment of the screening method. We also examined the expression of VLRB-pIII fusion protein of VLRB.HEL21 and VLRB.HEL2D phages by Western blotting. Crude lysates prepared from phages by heating the suspensions with reducing buffer were used for SDS-PAGE. VLRB-pIII was detected by mouse monoclonal anti-VLRB antibody, 4C4 [3]. Crude phage lysate from VLRB.HEL21 showed a 77 kD band matching the size of the VLRB-pIII fusion protein (arrowhead in Figure 2B). Owing to the crude nature of the phage lysates, additional bands are seen which are possibly attributed to spurious interactions between VLRB (and/or primary antibody) with other proteins as well as degradation of fusion proteins. However, VLRB.HEL2D phage showed a single band of expected size. To differentiate the VLRB-pIII fusion protein, bacterially expressed VLRB.HEL21 protein was included in this experiment. VLRB.HEL21 was cloned in pET28+ vector and His-tagged VLRB.HEL21 recombinant protein was isolated and differing amounts were run on the gel. Antibody 4C4 detected a 32 kD band corresponding to the size of the recombinant VLRB.HEL21. Therefore, the experiment clearly verifies the VLRB-pIII fusion protein in the phages and the presence of VLRB monomeric form expressed from *E. coli*.

### 2.2. Construction and Characterization of a Competitor Phage Display Library

The efficacy of the phage display screening for isolation of target specific lampribodies was tested by a subtraction experiment using a phage library of VLRB transcripts from unimmunized (naïve) sea lampreys spiked with a VLRB.HEL phage constructed above. As such, a VLRB library was constructed from the typhlosole (hematolymphoid organ) of two unimmunized lamprey larvae and characterized for its affinity for lysozyme as shown schematically in Figure 1A,B,D, representative PCR of the library showed diversity in size of the products (Figure 1C). Sequencing showed that the library consisted of full-length VLRB molecules with variable numbers of LRR cassettes and polymorphisms within them [26], and represented sufficient and non-redundant sequence diversity (detail analysis in Appendix A). The competitor phage library to be used for developing the screening method spiking with an anti-lysozyme VLRB driver phage will need to be devoid of any anti-lysozyme VLRB phage. Lysozyme binding of the library was examined by biopanning (described in Appendix A). After two rounds of biopanning, duplicate screening experiments were performed to isolate high affinity VLRB binders (Appendix A). No true binder was found; a few binders showing higher affinity in screening were further verified and were found to be false positives (Appendix A). This led to the conclusion that the library was devoid of any strong anti-lysozyme VLRB and could therefore be used for spiking with anti-lysozyme VLRB molecules for a subtraction experiment.

### 2.3. Subtraction Experiment for Recovery of the Driver Phage by Panning

In the subtraction experiment (described in Appendix A), the naïve phage library was spiked with the anti-lysozyme phage harboring VLRB.HEL21 which showed highest surface plasmon resonance (SPR) value [9]. In duplicate experiments, 1 PFU of VLRB.HEL21 phage was mixed with 10^9^ PFU of the library and the mixture was applied to lysozyme for biopanning. After two rounds of biopanning, the spiking phage, VLRB.HEL21, was recovered in both of the duplicate screening experiments (Appendix A) at percentages of 8.33 and 39.58 (Appendix A). This result demonstrates that the system is sensitive enough to recover a target specific VLRB from a pool of a billion such molecules at a significant quantity even when present in a single copy.

### 2.4. Immunization of Lamprey Larvae with Lysozyme and Isolation of Anti-Lysozyme VLRB Clones

Experiments were carried out to determine whether the VLRB phage display system was capable of isolating lampribodies from sea lampreys when challenged with a known immunogen, lysozyme. Three lamprey larvae were injected with KLH-conjugated lysozyme (see Materials and Methods). As a first step, the plasma from the immunized animals was examined for antigen binding by ELISA with biotinylated lysozyme. A ten-fold serial dilution of the plasma, 10^−1^ through 10^−12^, from each animal was used for ELISA with and without bait, and antibody 4C4 was used to detect lysozyme-bound VLRB molecules. The ELISA results demonstrated notable binding with the antigen at 10^−4^ dilution of the plasma from all three animals (Figure 3). These results, based on plasma reactivity, indicated that the immunization course yielded a good response.

In order to construct phage display libraries, polyA RNA isolated from the white blood cells of one of the immunized animals (animal 1) was used for cDNA synthesis. The VLRB specific primers, PD-VLRB.F and PD-VLRB.R (Appendix A) were used for PCR amplification of cDNA from animal 1. The amplicons were cloned into the phagemid vector pADL-10b to generate a library of full-length VLRB molecules, which were then propagated as phagemids. Efficiency of ligation was examined by PCR of 24 independent colonies with the same set of forward and reverse primers used for unimmunized animals (Appendix A). All the clones contained inserts of variable sizes, and the number of recombinants in the library was approximately 9.0 × 10^5^/μg insert DNA (~90,000 primary recombinants). Phages were generated from these phagemids and subsequently used for panning against biotinylated lysozyme on ELISA plates. Selective phage binding to the biotinylated lysozyme was monitored by calculating the percentage of recovery of the trapped phages using standard procedure for both phages bound to the bait and phages bound to the empty plate without bait (negative control). Nearly sevenfold more colonies were obtained from the bait-bound recovered phages suggesting the presence of target binding phages after the first panning. Bait-bound phages were recovered, amplified, and used for a second round of panning with and without bait. After recovering the phages, a small aliquot of infected *E. coli* was plated to obtain independent clones. Colonies were obtained only from the bait-bound recovered phages and no colonies were obtained from the negative control, suggesting significant selection of positive binders. Phagemids were isolated from 24 independent colonies and insert sizes were examined by PCR; all were positive for inserts of equal size (data not shown). Phages were prepared in batch from 12 of these colonies and screened for bait binding by ELISA and all were found to be positive. Inserts in these ELISA positive phagemids were sequenced. Among the 7 full-length VLRB sequences, 6 were identical (called Clone A) and another (called Clone B) differed by a single base at the 619th position (transition A to G) resulting in one residue change from threonine to alanine at the 207th position in the stalk region. Resequencing was not done for the remaining 5 phages as clonal expansion had already been observed in the 6 Clone A sequences. Fresh phages were prepared from one of the 6 identical clones for ELISA with lysozyme. The result (Figure 4) showed specific binding to lysozyme.

Alignment of the deduced protein sequence showed that the VLRB in the Clone A phage (Appendix A) was different from the three previously published VLRB.HEL clones and showed only 86–91% identity between them. Clone A showed no difference at the threonine residue from the other VLRB.HEL clones. Therefore, the single base transition seen in Clone B could be due to a de novo mutation from clone A or a naturally occurring variant and was selected because it did not compromise the binding affinity. Regardless, the fact that we saw clonal expansion after only two rounds of panning indicates that phage display is a viable method of screening. The fact that our isolated VLRB lampribodies showed only 86–91% identity to the previously isolated clones may be due, in part, to the fact that rearrangements are generated de novo in the lampreys and that the animals we use for our experiments are from an outbred source.

### 2.5. Isolation of VLRB Against Human IgM Protein

Human IgM protein crosslinked to paraformaldehyde-fixed Jurkat T cells was used for immunization of lamprey larvae (see Materials and Methods). The cDNA isolated from the leukocytes was used to amplify VLRB sequences with primers to the 5′ and 3′ untranslated regions (Appendix A). Approximately 100 ng of the PCR product thus amplified was re-amplified with PD-VLRB.F and PD-VLRB.R primers for 10 cycles (to avoid accumulation of nonspecifically primed and PCR mutated products). The amplicons were cloned into the pADL10b phagemid vector and a phage library was prepared. The number of recombinants in the library was approximately 1.6 × 10^9^/μL of insert DNA (1.6 × 10^8^ primary recombinants). The library was used for panning against human IgM, and phages recovered after the third round of panning were used to infect *E. coli*. Single colonies were obtained by plating an aliquot of the infected cells. PCR amplification of the inserts of 48 randomly isolated phagemids showed uniformity in the size of the inserts. Of these, 24 phages were screened for IgM binding by ELISA; 23 showed positive binding (data not shown). Of the ELISA positive phages, 10 were sequenced to determine if they had full-length VLRB molecules. The result showed that one was 810 bp and the remaining 9 clones were 807 bp (Appendix A); all were full-length VLRB. In the 9 VLRB phages of equal length (807 bp), there were two groups of identical clones, 2 in one group (called Clone A), 4 in another (called Clone B); the remaining 3 were non-clonal. Alignment of the DNA and the predicted protein sequences of the representatives of Clone A (10bIgM3P_4), Clone B (10bIgM3P_6), and the 4 unique clones (10bIgM3P_1, 10bIgM3P_3, 10bIgM3P_5, 10bIgM3P_2) revealed that the polymorphism found in all these clones were primarily confined to two locations, one within the connecting peptide (CP) region and the other in the C-terminal immediately upstream of reverse primer sequence (PD-VLRB.R) used for initial amplification of VLRB library. To determine the occurrence of the polymorphic bases, the sequences were compared against the non-redundant database in NCBI and mismatched bases were termed polymorphic. In case of Clone A (10bIgM3P_4) and another unique clone (10bIgM3P_3), the CP sequence was CATCTATCTGTT and in case of Clone B and two other clones (10bIgM3P_1 and 10bIgM3P_5), it was **G**ATCT**GG**CTGT**A** (the bolded bases are polymorphic in Appendix A). This short CP sequence in Clone A was found in the non-redundant database, but that of Clone B could not be found. At the 3′ of the stalk region, the polymorphic sequence in Clone A was **G**CT**A** and in Clones B and 10bIgM3P_3, it was TCTG. In this case, the sequences in Clone A and Clone 10bIgM3P_1 (**G**CT**A** and **C**CTG) were not found in the non-redundant database, but the other sequence was found. In another clone (10bIgM3P_5), the 3′ region showed a T to C transition, however, it was found in the non-redundant database. Clone 10bIgM3P_2 (810 bp) was the only one showing polymorphism in the LRRCT region and it was not found in the non-redundant database. The polymorphisms in the CP region and 3′ of the stalk region of the affinity selected molecules may or may not represent true variants as the cloning of the library involved two rounds of amplification by PCR. Appendix A shows the alignment of the deduced proteins sequence of the phages in S7A. Phages 10bIgM3P_1 (non-clonal), 10bIgM3P_2 (non-clonal), 10bIgM3P_4 (Clone A) and 10bIgM3P_6 (Clone B) were grown up to exponential phase and phages were prepared for ELISA with IgM, and the result showed comparable binding (Figure 5).

Clone A (VLRB.IgM3P_4) was introduced in phagemid vector pADL23c to isolate recombinant lampribody; this vector contains 6× His tag at the C-terminal for isolation of the recombinant protein and an amber mutation between VLRB and pIII to produce recombinant protein only instead of pIII fusion protein. Coomassie brilliant blue (CBB) stain of the SDS-PAGE shows that the recombinant protein expressed in *E. coli* was heavily contaminated with other proteins; a specific band of the estimated molecular weight of 32 kD could not be discerned after purification with Ni-NTA beads (Figure 6A). Similar results were obtained after cloning VLRB.IgM3P_4 into pET28+ vector and expressing in various host *E. coli* strains (data not shown). Further optimization of the culture and purification conditions for VLRB is underway. Western blotting with mouse monoclonal anti-VLRB antibody 4C4 showed the expected band in the purified fraction (Figure 6B). A smaller band of approximately 22 kD was found in the lysate after induction with IPTG (lane 1, Figure 6B), which is most likely a nonspecific protein binding to 4C4 antibody or a bacterially cleaved VLRB C-terminal fragment. This result suggested that the His-tag extract contained less than 10% of the actual recombinant protein. When expressed in insect cells Sf9, the protein was much less contaminated (data not shown).

Histological sections of human tonsil (lymphoid) tissue were subjected to IHC using the extract described above, which specifically stained a subset of cells, presumably IgM (Figure 7). The lampribody VLRB.IgM3P_4 has a cMyc tag in the C-terminal from pADL23c phagemid. Therefore, anti-cMyc antibody was used to detect VLRB.IgM3P_4. Positive staining of the IgM cells by VLRB.IgM3P_4 was also detected using 4C4 (data not shown). For the positive control, mouse IgG anti-human IgM antibody was used.

## 3. Discussion

Generation and isolation of antigen-specific lampribodies using phage display technology has been clearly demonstrated in the above experiments using lysozyme and human IgM as immunogens. The screening in phage display libraries relies on competitive binding and selection of the monomeric lampribody fused to the pIII molecule of the recombinant phage in each panning step. This allows for isolation of high affinity binders and their clonal expansion. By contrast, the screening method for isolating the binders in eukaryotic cells and yeast surface display does not allow competitive binding. Amplification of the relative high affinity binders in the panning rounds was observed in the case of both anti-lysozyme and anti-human IgM phage display screening. This was most likely due to the stringent screening conditions employed to eliminate selection of false positive clones.

By varying the stringency, the system should allow isolation of polyclonal lampribodies against an immunogen. It would also be interesting to map the epitopes in case of polyclonal responses. In a preliminary experiment, phages containing VLRB.HEL1, VLRB.HEL21, and VLRB.HEL2D molecules were mixed together with a näive VLRB phage display library at a ratio of 1 to one million. After the first round of panning under less stringent conditions, all three VLRB.HEL phages were recovered, emulating a polyclonal response. In the extreme dilution experiment described above, we used the same strategy to make serial dilutions of one of these (VLRB.HEL21) at a ratio of 1 to one billion and we were still able to faithfully recover the clone after only 3 rounds of selection. These results conclusively show that the VLRB sequences identified by another method (yeast surface display) can readily be selected when cloned in a phage display system.

Here, we have shown that the phage display system can be adopted to the variable lymphocyte receptors of the sea lamprey. The advantage of working with such a prokaryotic system is the ease at which the antibody repertoire can be cloned and screened. The reagents used for constructing and screening the library are off-the-shelf components and building these libraries is relatively easy and can be accomplished by most molecular genetics laboratories. The lampribodies so produced should be useful for most applications that conventional antibodies are used for and can easily be multimerized for use in certain cell biology applications. The lampribodies recognize both peptide epitopes [7] and glycotopes [13] and the phage display technology should prove useful as a screening tool when immunizing with more complex antigens, including whole cells. The key component in the generation of lampribodies is the immunization of the lampreys and capturing of its immune response as PCR amplicons. This technology should be portable to many laboratories that wish to generate a wide swath of reagents and is amenable to reagent sharing via an open source model.

## 4. Materials and Methods

### 4.1. Animals

Wild type sea lamprey larvae of 12–15 cm length were obtained from Acme Lamprey Company, Maine. Animals were maintained in partially sand filled aquariums at 16–18 °C and were fed on brewer’s yeast. All experiments were performed upon approval by the Institutional Animal Care and Use Committee at Western Fisheries Research Center, Seattle, Washington. For immunization by intraperitoneal injection, lamprey larvae were sedated in 0.1 g/L of MS222 (tricaine methanesulfonate, Sigma, St. Louis, MO, USA) buffered with sodium bicarbonate. The animals were euthanized by immersion into 1 g/L of MS222 buffered with sodium bicarbonate for collection of blood and organs.

### 4.2. Immunizations

Sea lamprey larvae were immunized as described in [6] with minor modification. Lysozyme from chicken egg white (Sigma) was conjugated with keyhole limpet hemocyanin (KLH) using Mariculture KLH and EDC Conjugation Kit (Fischer Scientific, Lenexa, KS, USA) according to the manufacturer’s instructions. The conjugation reaction mixture was dialyzed against phosphate-buffered saline (PBS) overnight at 4 °C. Concentration of the conjugated lysozyme was measured by Pierce BCA Protein Assay Kit (Thermo Scientific, Fresno, CA, USA), divided into aliquots, and stored at −80 °C. For each immunization, 70 µg of the conjugated lysozyme in 30 μL volume mixed with equal volume of Imject Alum (Fischer Scientific) was used according to manufacturer’s instructions. Three doses of immunogen were injected intraperitoneally every 2 weeks at 4, 6, and 8 weeks. Each animal was euthanized two weeks after the final injection. When signs of life were undetectable, 100 μL of PBS with 30 mM EDTA were injected into the peritoneal cavity to generate intracoelomic pressure. The tail was immediately severed 1 mm proximal to the cloacal opening. Blood was collected in a tube containing 0.5 mL of PBS with 30 mM EDTA, mixed well by inverting and immediately centrifuged at 106× *g* for 2 min. Plasma was separated and stored at −80 °C and the cells were resuspended in 0.5 mL of PBS with 30 mM EDTA and kept on ice until the isolation of buffy coat. The animal was dissected by ventral midline incision, and the kidneys, intestine and typhlosole were harvested, snap frozen, and stored at −80 °C. To isolate the buffy, the cell suspension was gently pipetted into a 2 mL microfuge tube containing 1 mL of 55% Percoll (GE Healthcare, San Ramon, CA, USA) and centrifuged at 4 °C at 400× *g* for 5 min in a swing bucket rotor with minimal setting of speed ramp and brake. The buffy coat was isolated (0.2 to 0.3 mL) and mixed with 0.75 mL TRIzol reagent (Invitrogen, Waltham, MA, USA) and stored at −80 °C until further processing. Blood and organs from unimmunized animals were isolated using the same procedure.

All experiments of immunization of sea lamprey larvae (obtained from Lamprey Services, Ludington, MI) with human IgM protein were done at Emory University with the approval of the Emory University Institutional Animal Care and Use Committee (IACUC). For immunization, purified human IgM from plasma (EMD Millipore, Temecula, CA, USA) was coupled to fixed Jurkat T cells as an adjuvant. 10^8^ Jurkat T cells were fixed overnight in 4% paraformaldehyde, then washed in 20 mM MES, pH 5.5, and activated with EDC/NHS (1-ethyl-3-(-3-dimethylaminopropyl) carbodiimide hydrochloride/N-hydroxysuccinimide) for amine conjugation for 20 min at room temperature. The cells were briefly washed in PBS, then 0.5 mg of human IgM in 1 mL of PBS was added to the pelleted EDC/NHS-activated cells for 3 h at room temperature on a rotating mixer. At the end of the conjugation reaction, the cells were centrifuged at 400× *g* and the supernatant was removed, then the cells were washed in PBS with 10 mM Tris, pH 7.5. The IgM conjugated Jurkat cells were stored at 4 °C until needed for immunization. Lamprey larvae were sedated with 0.1 g/L tricaine methanesulfonate (Tricaine-S; Western Chemical, Inc., Riverside, CA, USA), before injection with 10^7^ IgM conjugated Jurkat cells resuspended in 30 µL of PBS, such that each lamprey received approximate 50 µg of recombinant IgM using the same immunization schedule as that for lysozyme.

### 4.3. Naive VLRB Phage Display Library Construction

Two unimmunized animals were used for this purpose. From each, total RNA was extracted from 20 mg typhlosole using TRIzol reagent (Invitrogen, Waltham, MA, USA). From the total RNA, polyA RNA was isolated using NucleoTrap mRNA Mini Kit (Clontech, Mountain View, CA, USA). An aliquot of the polyA RNA was used for cDNA synthesis using SuperScript III (Invitrogen). Entire VLRB molecule including the hydrophobic tail was amplified from the typhlosole cDNA PD-VLRB.F and PD-VLRB.R (Appendix A). The forward primer contains a 6 basepair insert, atggcg, to obtain complete PelB leader encoding sequence. Both forward and reverse primers contain *Sfi*I*/Bgl*I overhang for directional cloning of VLRB inserts in pADL phagemid (Antibody Design Laboratories, San Diego, CA, USA). Agarose gel electrophoresis of the PCR products showed a size range of 0.8 to 1.2 kb. The PCR products were digested by *Bgl*I restriction enzyme and 50 ng of the digested product was ligated to 25 ng *Bgl*I digested and dephosphorylated phagemid vector (pADL) overnight at 16 °C (Figure 1C). The reaction mixture was extracted twice with phenol: CIAA, precipitated in ethanol and resuspended in 10 μL of water. From this, 2 μL were used to transform electrocompetent TG1 *E. coli*. The cells were recovered in 1 mL of 2× YT supplemented with glucose and thiamine-HCl to a final concentration of 2% and 1 mM, respectively, and grown for 1 h at 37 °C with shaking at 250 rpm. After the incubation, 10 μL of the culture was taken out and plated for determination of independent clones. The rest of the cells were grown for 6 h in 2× YT containing 2% glucose and 1 mM thiamine-HCl and 100 μg/mL carbenicillin to prepare the primary library. Half of this library was used to prepare phage and the remaining half was stored at −80 °C adding 20% glycerol. Similarly, phagemid library was constructed from the buffy coat of animals immunized by lysozyme. For a VLRB library against human IgM, RNA was isolated from the buffy coat and reverse transcribed into cDNA with oligo-dT priming. VLRB transcripts were amplified from the cDNA by PCR using primers specific for the 5′ and 3′ untranslated region (Appendix A).

### 4.4. Subcloning of Anti-Hen Egg Lysozyme (HEL) VLRB Molecules for Phage Display

Diversity region mRNA of VLRB.HEL.21, VLRB.HEL.1, and VLRB.HEL.2D (GenBank accessions FJ794807, FJ794806, and FJ794805 respectively) were extracted (Figure 1A). Full-length VLRB sequences were constructed in silico by adding conserved 5′ and 3′ sequences to the diversity regions gene fragments and the constructs were synthesized (Life Technologies, South San Francisco, CA, USA) with *Bgl*I restriction enzyme sites for directional cloning into phagemid vectors, pADL10b or pADL23c.

### 4.5. Phage Preparation from the VLRB Phage Display Libraries and the VLRB HEL Phage Display Clones

To 0.5 mL of the primary library, 9.4 mL of 2× YT containing 2% glucose and, 1 mM thiamine-HCl, 100 μg/mL carbenicillin and 100 μL of 10^10^ PFU of Hyperphage (M13K07ΔpIII, Progen Biotechnik GmbH, Germany) were added and the culture was grown for 1 h at 37 °C with shaking at 250 rpm. The culture was centrifuged at 2000× *g* for 10 min at 4 °C and supernatant was discarded as much as possible to get rid of the traces of glucose. The cells were resuspended in 10 mL of 2× YT medium and carbenicillin, kanamycin was added to a final concentration of 100 μg/mL, and IPTG to 250 μM and incubated at 30 °C for 20 h. The culture was centrifuged at 10,000× *g* for 20 min at 4 °C, the supernatant containing recombinant phages was collected. To precipitate recombinant phages, first equal volume (10 mL) of blocking buffer (PBS with 0.05% Tween 20 and 1% BSA) was added to the recombinant phages and incubated for 10 min on ice; then 20% polyethylene glycol was added to a final concentration of 3.33% and incubated on ice for 1 h. It was then centrifuged at 10,000× *g* for 20 min at 4 °C. The supernatant was discarded, and the pellet was resuspended in 0.4 mL of ice-cold PBS. The phage solution was transferred in a sterile microfuge tube and centrifuged at 14,000 rpm for 10 min at 4 °C. The supernatant was saved in a sterile microfuge tube and stored at 4 °C. For, VLRB.HEL phagemid clones, single colony was isolated from each of the TG1 cells transformed with VLRB.HEL phagemid and grown in broth up to exponential phase and 1 mL of the culture was used to prepare phage as described above.

### 4.6. Titering the VLRB Phage Display Libraries and Individual VLRB.HEL Phage

TG1 cells from glycerol stock were spread on 2× YT agar plate with 2% glucose and 1 mM thiamine-HCl (2× YTGT). Single colony was isolated from overnight plate culture and used to inoculate 10 mL of 2×YT liquid broth with 2% glucose and 1 mM thiamine-HCl and incubated at 37 °C with shaking at 250 rpm. The cells were grown until the optical density (OD) at 600 nm reached 0.4–0.6. The culture was immediately chilled on ice slurry and kept on ice until use. Ten-fold serial dilutions of the phage solution of the library was made in 2× YT medium. To 100 μL of 10^−6^, 10^−7^, 10^−8^ dilutions of the phage solution, 100 μL of pre-chilled exponential-phase TG1 cells were added. After gentle mixing, incubation was continued at room temperature for 15 min. Each dilution was plated on to 2 plates and incubated overnight at 37 °C. Plates having 30 to 300 colonies were counted and the titer of the phage solution was determined. Titer of individual VLRB.HEL phage solution was determined in the same way.

### 4.7. Biotinylation of the Bait and ELISA Screening

Lysozyme from chicken egg white (Sigma, Burlington, MA, USA) was dissolved in water to a final concentration of 10 mg/mL, 0.2 mg of the solution was labeled with biotin using EZ-Link Micro NSH-PEG_4_—Biotinylation Kit (Thermo Scientific, Fresno, CA, USA) according to manufacturer’s instructions. Biotinylation was estimated using Pierce Biotin Quantitation Kit (Thermo Scientific) and lysozyme concentration after labeling was measured by Pierce BCA Protein Assay Kit (Thermo Scientific). Pierce Streptavidin High Binding Capacity Coated 96-Well Plates (Thermo Scientific) were washed three times with wash buffer (1× PBS with 0.05% Tween 20). Biotinylated lysozyme was diluted to 0.5 ng/μL with 1× PBS and 100 μL were used for coating each well (50 ng per well). The plates were incubated at 37 °C for 1 h or 4 °C for a minimum of 4 h on Maxi Rotor (Lab Line) with shaking at 10 rpm. The same setting of the rotor was used throughout the experiment. The bait solution was removed, and the wells were washed three times with wash buffer. Blocking buffer (200 μL) was added to each well, and incubated at 37 °C for 1 h or 4 °C for a minimum of 4 h. The blocking buffer was removed, and the wells were washed three times with wash buffer. To each well, 50 μL of phage solution mixed with equal volume of blocking buffer was added and incubated at 37 °C for 1 h or 4 °C for a minimum of 4 h. In the case of plasma from immunized animals, 50 μL of dilute plasma were mixed with equal volume of blocking buffer and added to the well. The phage solution was removed, and the plate was washed 5 times with wash buffer. HRP/anti-M13 Monoclonal Conjugate (GE Healthcare) was diluted 1:2500 in blocking buffer and 100 μL was added to each well and incubated at 37 °C for 1 h or 4 °C for a minimum of 4 h. The antibody was removed, and the wells were washed 5 times with wash buffer. Equal volume of peroxidase substrate (TMB) and peroxidase solution (Thermo Scientific) was mixed and 100 μL were added to each well and incubated at room temperature for 20 min, until a suitable blue color developed. The reaction was stopped by adding 100 μL of 2M sulfuric acid. The absorbance of each well was measured at 450 nm using OptiMax Tunable Microplate Reader (Molecular Devices). In case of plasma from immunized animals, VLRB antibody 4C4 (6) was used as the primary antibody and it was further detected by peroxidase-conjugated rabbit anti-mouse IgG in 1:500 dilution (Jackson ImmunoResearch Laboratories, Inc., West Grove, PA, USA)

### 4.8. Panning, Rescue, and Superinfection

Panning was performed as shown schematically in Appendix A. First round of panning: Biotinylated bait (lysozyme) coated plates were prepared as described above. For panning, 10^9^ phage PFUs were used. The solution was prepared by mixing 50 μL of the phage containing a maximum of 10^9^ phage PFUs with 50 μL of blocking buffer. At first the total 100 μL phage suspension was applied to an empty well (without coating with biotinylated lysozyme) and incubated at 37 °C for 30 min. This pretreatment step was done to eliminate non-specific phages. After that the phage suspension was recovered by pipetting and transferred to a biotinylated lysozyme coated well. The plate was incubated 37 °C for 1 h or 4 °C for a minimum of 4 h. The unbound phage suspension was removed, and the wells were washed 6 times with wash buffer. To each well, 100 μL of 0.2 M glycine-HCl, pH 2.2, was added and the plate was incubated at room temperature for 15 min to release the bound phages. In a microfuge tube, 100 μL of 0.2 M Tris-HCl, pH 8.0 were added and kept on ice. After 15 min, the glycine-HCl solution was pipetted out of the well and added to the microfuge tube containing Tris-HCl solution to neutralize the solution, the total volume of the phage solution became 200 μL. TG1 cells were grown to exponential phase from a single colony, as described above. The culture was divided into 2 mL aliquots in 15 mL culture tubes and 200 μL of released phase solution was added to each. The tube was incubated at 37 °C for 30 min without shaking and another 30 min with shaking at 250 rpm. The culture was centrifuged at 2000× *g* for 10 min at 4 °C and supernatant was discarded as much as possible to get rid of the traces of glucose. The cells were resuspended in 100 μL of 2× YTGT broth, representing the enriched phage library. From this, 10^−1^, 10^−2^, and 10^−3^ dilutions were prepared in 100 μL of 2× YTGT medium and plated in 2× YTGT carbenicillin (100 μg/mL) agar plates and incubated at 37 °C overnight. The colonies were counted, and the percent of phage recovery was calculated. To prepare phage, 1 mL of 2× YTGT broth was added to 50 μL of cells suspension from enriched phage library and incubated at 37 °C for 1 h. Then, 100 μg/mL carbenicillin and 5 × 10^9^ PFU of Hyperphage were added and the incubation was continued at 37 °C for 1 h with shaking at 250 rpm. The culture was centrifuged at 2000× *g* for 10 min at 4 °C and supernatant was discarded as much as possible to get rid of the traces of glucose. The cells were resuspended in 10 mL of 2× YT medium and phage preparation was continued as described above. The rest of the enriched phage library (approximately 40 μL were stored in 10% glycerol at −80 °C. Phage solutions were stored at 4 °C. Panning was repeated as described above using 50 μL of the phage solution from the first or previous round of panning.

### 4.9. ELISA Test of the Candidate Phage Clones

The colonies obtained in percent of phage recovery experiment were used for ELISA. Single colonies were streaked on 2× YTGT carbenicillin (100 μg/mL) agar plates and also grown in 300 μL of 2× YTGT carbenicillin (100 μg/mL) broth in microfuge tubes. The plates were incubated at 37 °C overnight and the cells were used for PCR and phagemid isolation later. An individual candidate phage clone from a single colony used for phage preparation was named Phage Clone 1, or PC1 when TG1 contained the recombinant phagemid; the same clone was named PC2 when phage adding helper phage to the culture; finally, it was named PC3 when phage was resuspended. An aliquot of PC1 cultures was stored in −80 °C as the primary culture. In addition, individual candidate phage clones were also assigned unique identity. This combined designation was useful for organization of the experiment. The microfuge tubes (labeled PC1 and the clone ID) were incubated at 37 °C overnight. From these cultures, 50 μL were transferred to a new microfuge tube labeled PC2 and the clone ID containing 300 μL of 2× YTGT broth with 100 μg/mL carbenicillin and 5 × 10^8^ Hyperphage. Incubation was carried out at 37 °C for 1 h. The remaining culture was stored in 10% glycerol at −80 °C. The cells in the culture were collected by centrifugation at 2000× *g* for 3 min at 4 °C. The supernatant was discarded, and the cells were resuspended in 350 μL of 2× YTGT with carbenicillin, kanamycin added to a final concentration of 100 μg/mL, and IPTG to 250 μM, followed by incubation at 30 °C overnight with shaking. The microfuge tubes were centrifuged at 14,000 rpm for 10 min at 4 °C and 300 μL of the supernatant containing the phage were transferred to a new microfuge tube labeled PC3 and the clone ID. To this, 300 μL of blocking buffer (for phage precipitation) were added and incubated on ice for 10 min. Then 120 μL of 20% PEG were added to each tube and incubation was continued on ice for 1 h. The microfuge tube was centrifuged at 14,000 rpm for 20 min at 4 °C. The supernatant was discarded, and the phage PFUs were resuspended in 100 μL of ice-cold PBS. From the phage solution 50 μL were mixed with equal volume of blocking buffer and applied to biotinylated lysozyme coated well for ELISA as described above.

### 4.10. PCR, Enzyme Digestion and Sequence Analysis of the Phagemid Clones

The PC1 clones streaked on plates were used for screening by PCR and enzyme digestion and isolation of phagemids for sequencing. PCR was performed in a 10 μL reaction volume using a forward primer, CAPBSF1 (5′-ATGTGAGTTAGCTCACTCATTAGGC-3′), located upstream, and a reverse primer, GPIIICTR2 (5′-TGTCGTCTTTCCAGACGTTAGTAAATG-3′), located downstream of *Bgl*I cloning sites in the vector. The phagemid with any insert produced a fragment of 329 bp. The PCR product was checked by agarose gel electrophoresis. In the case of screening for the VLRB.HEL clones, the PCR product was digested with *Bgl*I and *EcoR*I to produce a small fragment (684 bp for VLRB.HEL.21) which allowed screening for VLRB.HEL.21 clones. The candidates were sequenced from DNA Sequencing Service of Operon, Fischer Scientific. Sequence contig assembly was done by Vector NTI program (Invitrogen) and alignment was done by ClustalW and ClustalX version 2 [27].

### 4.11. Isolation of Bacterially Expressed VLRB Recombinant Proteins, Western Blotting, and Immunohistochemistry (IHC)

VLRB.HEL21 was cloned in pET28+ vector for expression of His-tagged protein in *E. coli* BL21. BL21 containing the recombinant plasmid was grown in 2× YT medium with 10 μg/mL of kanamycin in a shaker at 250 rpm at 37 °C. When the OD reached 0.6, IPTG was added at a concentration of 1 mM/L and incubation was continued overnight at 30 °C. The cells were harvested by centrifugation at 4 °C, washed twice with ice-cold PBS, and finally resuspended in His-tag lysis buffer, 40 mL/L of culture. It was incubated on ice for 30 min. Lysozyme was added at a concentration of 50 μg/mL and RNaseA at 10 μg/mL and incubation was continued for another 30 min. The cell lysate was sonicated on ice for 10 cycles, each with a 10 s pulse at 70% output and 30 s intervals. It was incubated on ice for 10 min and then homogenized thoroughly, and again incubated on ice for 10 min. The process was repeated twice more. For a freeze thaw cycle, the lysate was dipped into liquid nitrogen until the entire content solidified and then incubated in 37 °C for 30 min. Then, the lysate was left at room temperature and homogenized every 10 min. The lysate was centrifuged at 9000 rpm at 4 °C for 30 min. The supernatant was transferred in a new tube and again centrifuged for 1 h. The supernatant was transferred to a new tube and 2 mL of His-tag lysate buffer equilibrated Ni-NTA agarose beads was added and incubated at 4 °C for 2 h. The lysate was poured into a high flow column. The column was washed with 50 mL of wash buffer and eluted with 1.0 mL aliquots of elution buffer. Concentration of the eluted protein was measured by BCA method.

The eluted fractions were electrophoresed in 10% SDS-PAGE (Lonza, Hayward, CA, USA), stained with Coomassie brilliant blue solution (G-Biosciences, St. Louis, MO, USA), and destained with an aqueous solution of 40% methanol and 10% glacial acetic acid. The eluted fractions having higher concentration were pooled; however, the recombinant protein had many other bands associated with it after His-tag purification. For Western blotting, SDS-PAGE was run, transferred (semi-dry) to a nitrocellulose membrane using iBlot (Invitrogen) system. The blot was blocked with 5% blotto in TBST (150 mM NaCl, 10 mM Tris, 0.1% Tween 20) for an hour at room temperature. It was incubated with primary antibody, 4C4, at 1 μg/mL of blocking buffer overnight at 4 °C. The blot was washed thoroughly with TBST and incubated with horseradish peroxidase (HRP) conjugated rabbit anti-mouse IgG secondary antibody at 1:1000 dilution (Jackson ImmunoResearch Laboratories, Inc., West Grove, PA, USA) for 2 h at room temperature. The blot was washed and a substrate solution (Luminata Forte, Millipore, Temecula, CA, USA) was added. Image was obtained immediately using ChemiDoc Imaging System (Bio-Rad, Hercules, CA, USA).

VLRB.IgM3P_4 was cloned in pADL23c phagemid for expression in *E. coli* BL21. A single colony was isolated from *E. coli* containing the phagemid and was grown in 10 mL of LB liquid medium with 100 μg/mL of carbenicillin in a shaker at 250 rpm at 37 °C. After overnight incubation, the culture was diluted in 1 L of LB medium supplemented with 2% glucose, and 100 μg/mL of carbenicillin was added; incubation was continued, and optical density was measured at 600 nm every hour. At OD 0.6, cells were collected by centrifugation and resuspended in M9 liquid medium supplemented with 2 mM MgSO_4_, 0.1 mM CaCl_2_, 0.4% glucose, and 1 mM thiamine-HCl. Carbenicillin was added at a concentration of 50 μg/mL and the culture was incubated at 16 °C with shaking for an hour for cells to acclimatize. Then, IPTG was added at a concentration of 500 nm/L and incubation was continued overnight. Cell lysate preparation, isolation of His-tagged protein, and Western blotting were performed as described above. As in VLRB.HEL21, the eluted fractions had many other bands associated with it after His-tag purification of VLRB.IgM3P_4. Similar result was obtained when VLRB.IgM3P_4 was cloned into pET28+ vector and expressed. The protein was dialyzed with 10 kD cutoff spin column to remove salts. Then, the protein was again purified using the Ni-NTA bead and was used for IHC.

IHC was performed using standard protocol [8]. Human tonsil sections were placed on a slide and deparaffinized by three changes in xylene, 10 min each. The sections were rehydrated in serial dilution of 100%, 95%, and 80% ethanol and finally in water. Heat-induced epitope retrieval (HIER) was done in Na-citrate buffer (10 mM sodium citrate, pH 6.0) in a steamer for 15 min and cooled to room temperature for 10 min. The slides were washed and immersed in PBST (PBS with 0.1% Triton × 100). To block endogenous peroxidase, the slides were incubated with 3% aqueous solution of hydrogen peroxide at room temperature and then washed with PBST again. The slides were perfused in PBST with BSA (1%) blocking buffer for an hour at room temperature and then incubated with 5 μg of the polled His-tag purified recombinant protein, VLRB.IgM3P_4 in 100 μL of blocking buffer (PBST with BSA) at 4 °C overnight. The slides were washed with PBST buffer and then incubated with anti-cMyc antibody at 1:100 dilution (tag in the C-terminal of recombinant pADL23c phagemid, from Thermo Fisher Scientific), under the same condition. The slides were washed with PBST and incubated with HRP conjugated mouse secondary antibody at 1:500 dilution (Jackson ImmunoResearch Laboratories, Inc.). Antigen–antibody complexes were visualized by DAB peroxidase substrate and counterstained with Meyers hematoxylin (Vector Laboratories, Burlingame, CA, USA). VLRB.IgM3P_4 was also detected with 4C4 at 1 μg/100 μL of blocking buffer. For the positive control, mouse anti-human IgM antibody at 1:100 dilution (Biolegend, San Diego, CA, USA) was used.

## Figures and Tables

**Figure 1 biomolecules-09-00868-f001:**
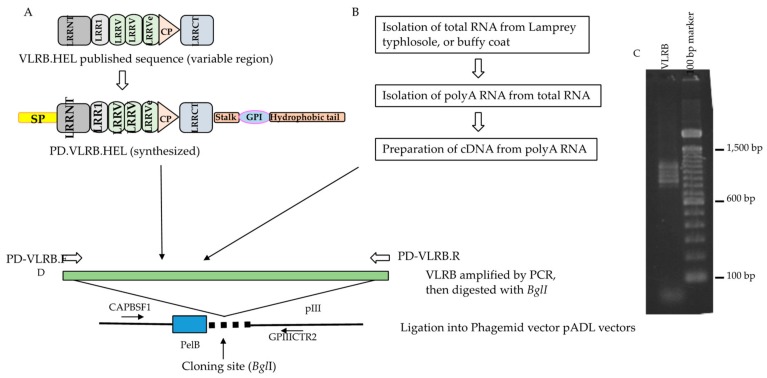
Cloning of anti-lysozyme VLRB, VLRB.HEL in a phagemid vector and construction strategy of VLRB library. (**A**) Three full-length anti-HEL VLRBs, diversity region mRNA, partial coding sequences were extracted from GenBank and conserved regions were added to the 5′ and 3′ ends to construct full-length molecules. VLRB.HELs were then synthesized and amplified by PD-VLRB.F (forward) and PD-VLRB.R (reverse) primers for directional cloning into the phagemid vector. (**B**) Unimmunized (also for immunized) lamprey larvae were euthanized, and organs harvested. Total RNA and then polyA RNA was isolated from the typhlosole of unimmunized (buffy coat of immunized animals). PolyA RNA was converted to cDNA which was used as a template for preparation of VLRB amplicon library by PCR with PD-VLRB.F and PD-VLRB.R primers. (**C**) Agarose gel (1.5%) electrophoresis of VLRB amplicons from typhlosole from unimmunized animals mentioned above. A ladder pattern ranging from 0.8 to 1.2 kb was seen. (**D**) The VLRB amplicon libraries from the typhlosole and each of the VLRB.HELs were digested with *Bgl*I and directionally cloned in phagemid vectors, pADL10b or pADL23c. The sequence of the insert was determined using a forward primer (CAPBSF1) and a reverse primer (GPIIICTR2) located upstream and downstream of the cloning site respectively. Approximately 70% of the single colonies had inserts ranging from 0.5 to more than 1.0 kb, the remaining colonies either possessing small inserts or no inserts. The number of recombinants in the library was approximately 7 × 10^5^ (7.0 × 10^6^ cfu/μg of insert DNA). A small number of the phagemid clones with estimated inserts ≥0.7 kb was sequenced to see if the library had full-length VLRB, and to assess their sequence diversity. Sequences for 27 clones were obtained, all of which were full length and encompassed unique VLRBs. Their putative protein sequences showed extensive homology with the reported VLRBs in the non-redundant protein sequences database (data not shown).

**Figure 2 biomolecules-09-00868-f002:**
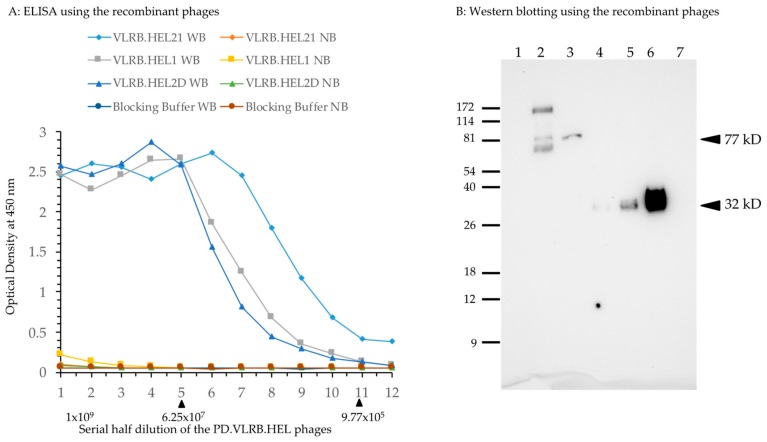
VLRB.HEL recombinant M13 phage preparation and ELISA test for its binding to lysozyme. (**A**) Each well of an ELISA plate was coated with 50 ng of biotinylated hen egg lysozyme to which 100 μL of blocking buffer was added. To prepare two-fold serial dilutions of the phage, 100 μL of phage (2 × 10^9^ PFU) was transferred to the first well, mixed, and then 100 μL solution from the first well was transferred to the second well and mixed. The series of two-fold serial dilutions was continued for the rest of wells in the same row, with and without the bait. WB: with bait; NB: no bait in the wells in a particular row. ELISA was carried out using standard procedures. Optical density at 450 nm was plotted against the number of phage PFUs per respective well. Blocking buffer alone was used to assess background binding of the phages as well as for control experiments employing the primary antibody only (no phages). The arrowhead on the right underneath the X-axis marks the phage concentration at which the binding became appreciable; that on the left marks the phage concentration at which the binding reached a plateau for all the phages. Numbers in X-axis denotes serial half dilution of the phage solutions from 1 × 10^9^ to 4.88 × 10^5^ PFU. (**B**) Phages prepared from VLRB.HEL21 and VLRB.HEL2D were examined for the expression of VLRB-pIII fusion protein. Approximately 4 × 10^9^ PFUs were heated to 95 °C for 5 min in reducing buffer and run in an SDS-PAGE gel. His-tagged VLRB.HEL21 protein was isolated by cloning the full-length fragment into pET28+ vector and expressed in *E. coli* BL21. Serial concentration of His-tagged eluate of VLRB.HEL21 was used to demonstrate the presence of the protein. Lane assignments: 1, ProSieve Color Protein Marker in kD; 2, VLRB.HEL21 phage; 3, VLRB.HEL2D phage; 4–6, His-tagged VLRB.HEL21 protein 1, 2, and 4 μg respectively; 7. Zebrafish protein, 5 μg (used as a negative control. Arrowheads indicate the position of the bands with estimated molecular weights of VLRB-pIII fusion protein and bacterially expressed His-tagged VLRB.HEL21 protein in the blot.

**Figure 3 biomolecules-09-00868-f003:**
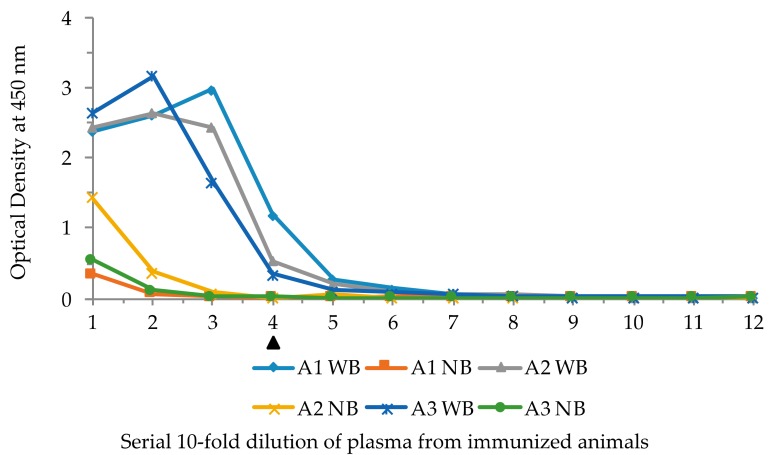
ELISA results of the plasma from three lysozyme-immunized sea lampreys showing reactivity against biotinylated bait and no bait. Serial ten-fold dilutions of the plasma were added to wells with and without the bait. A1, A2, and A3 are the individual animals, WB: with bait, NB: with no bait in the wells. Solid triangle underneath the *X-*axis marks the lowest limit of appreciable OD. Numbers in *X-*axis denotes serial half dilution of the phage solutions from 10^−1^ to 10^−12^. WB denotes with bait, and NB no bait in the wells in that row.

**Figure 4 biomolecules-09-00868-f004:**
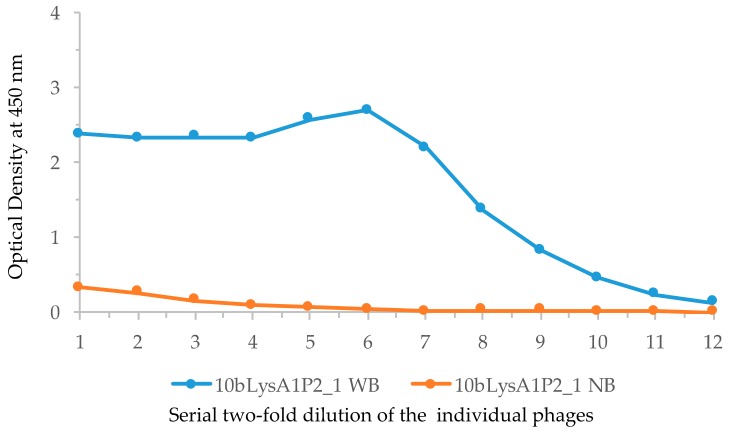
Confirmation of lysozyme binding by purified clone from the screening experiment. Lysozyme binding of one VLRB clone isolated from the immunized animal (A1), 10bLysA1P2_1 was examined by making phage from log phase culture by ELISA. Serial half dilutions beginning with 10^9^ PFU were added to wells in a microtiter plate coated with biotinylated lysozyme. Wells without biotinylated lysozyme were used as negative controls. WB, with bait; NB, no bait in the wells. Numbers in *X-*axis denotes serial half dilution of the phage solutions from 1 × 10^9^ to 4.88 × 10^5^ PFU.

**Figure 5 biomolecules-09-00868-f005:**
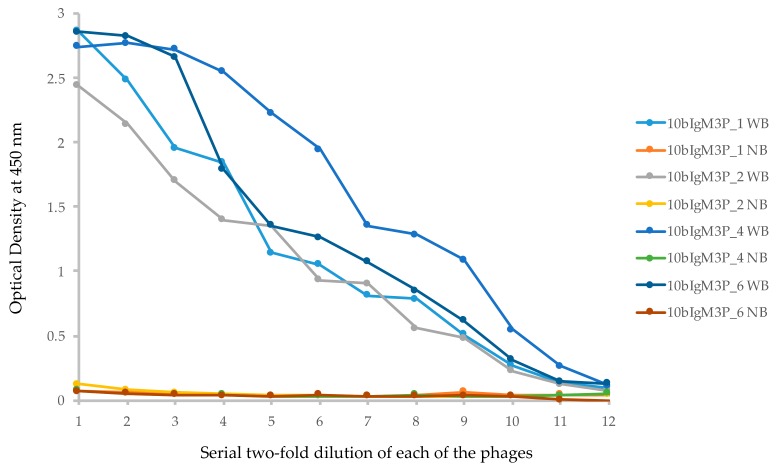
ELISA result of the four non-redundant VLRB clones against human IgM isolated by phage display screening. Each well of an ELISA plate was coated with 75 ng of biotinylated human IgM. Serial two-fold dilution of 10^10^ PFU of each of the phages was used. Each sample was used with and without bait. WB: with bait; NB: no bait in the wells. Numbers in *X-*axis denotes serial half-dilution of the phage solutions from 1 × 10^9^ to 4.88 × 10^5^ PFU.

**Figure 6 biomolecules-09-00868-f006:**
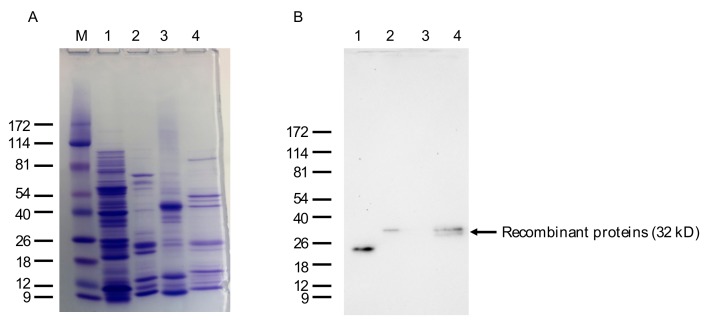
SDS-PAGE and Western blot analysis of the bacterially expressed anti-IgM VLRB lampribody. (**A**) Approximately 2 mg of pooled His-tag eluate of VLRB.IgM3P_4 was run and stained with Coomassie brilliant blue (CBB). Lanes: M, ProSieve Color Protein Marker in kD; 1, 5 mg of cell lysate after induction with IPTG; 2, 2 mg of pooled VLRB.IgM3P_4 protein; 3, 5 mg of Zebrafish muscle protein for comparison; 4, 2 mg of His-tagged VLRB.HEL21 protein. (**B**) An identical gel was run for Western blotting. Zebrafish protein and and VLRB.HEL21 protein were used as negative and positive controls for the primary antibody, 4C4.

**Figure 7 biomolecules-09-00868-f007:**
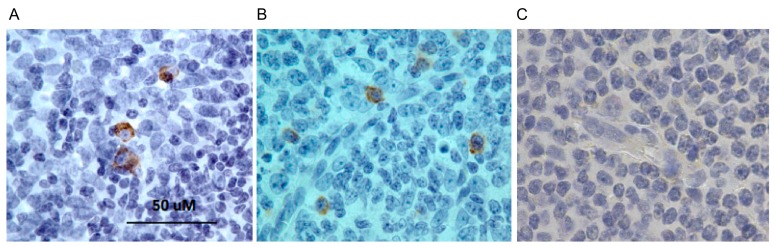
Immunohistochemistry using anti-IgM lampribody. Histological sections of human tonsil (lymphoid) tissue were stained with (**A**) lampribody, VLRB.IgM3P_4 or (**B**) mouse IgG anti-human IgM antibody. The detection of the VLRB was done using anti-cMyc antibody. The final detection was done with HRP-conjugated rabbit anti-mouse IgG secondary and using diaminobenzidine (DAB) substrate. Cells reacting positively with the lampribody (**A**) and the IgG reagent (**B**) were stained dark brown. (**C**) Negative control was stained similarly except without any primary antibody.

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
