# Peer review of "Generation of Lamprey Monoclonal Antibodies (Lampribodies) Using the Phage Display System"

_biomolecules, 2019, doi:10.3390/biom9120868_

Round 1

Reviewer 1 Report

The authors have properly responded the questions or concerns being raised on a point-to-point basis. However, a few revisions or comments need to be refined or respond before the manuscript is ready for publication.

In figure 1C, the authors need to verify the amounts (in mg or mg?) of His-tagged VLRB.HEL21 protein loaded onto the SDS-PAGE. It is strongly suggested that the positive clone isolated previously from sea lamprey larvae be included in figure 2 in revised manuscript for comparison. Answer to question #6 is not comprehensive. It is well established that PCR is commonly used to examine the clones with the right size of antibody gene inserts; however, the result only do not reflect the real clonality of the randomly selected clones after each panning. It is believed that the clonality analyzed from the 2nd and 4th panning steps could be different. Thus, it is more meaningful if the antibody gene sequences of the selected clones from different panning steps could be analyzed and presented for comparison. The data in figure 5 need to be presented in a more appreciated way. It is suggested that the authors simultaneously run 2 SDS-PAGE containing 5 identical samples of protein markers, cell lysates after induction, purified VLRB.IgM3P_4, purified VLRB.CT and Zebrafish muscle protein (negative control). One gel is for CBB staining to show the protein profiles; the other for western detection to show the purified VLRB.IgM3P_4. In addition, it is not reasonable to see plenty of protein bands in lane 3 in figure 5A. It is also confusing to detect the same protein patterns with different intensity in lanes 3 and 4 in figure 5B. More specifically speaking, why recombinant VLRB.CT was also detected in the preparation of Zebrafish muscle protein in lane 3?

Reviewer 2 Report

Abstract

Line 29: “…high affinity binders”. Did the Authors evaluate affinity of the binders? Probably, it will be more accurate to talk about binders with relative high affinities as identified by competitive binding. Please clarify.

Line 31: “… highest affinity binders”. See the comment above.

Introduction

Line 73: Please verify/clarify references 23 and 24.

Lines 76-77: “In this study, we evaluated the robustness of the phage display system for cloning and screening VLRB specificities”. Please state objectives of the study clearly (objective 1 was …..objective 2 was …).

Results

No reference to Figure 1A in the Results section.

Figure 1A: This figure does not provide any new information. Please remove.

Figure S1 “Cloning of anti-lysozyme VLRB, VLRB.HEL in a phagemid vector and construction strategy of VLRB library” has to be included in the main text since construction of the library is one of the objectives of the study.

Lines 96-97: “To test the binding affinity….” Instead, it might be more accurate to talk about “relative binding affinity”.

Line 106: “… slight differences in the binding affinity…” Please define what “slight” means.

Line 109: “Crude lysates prepared from phages by heating the solutions with reducing buffer…” It is more accurate to say phage suspensions, not solutions.

Line 113: “…. contain additional bands that may have resulted from clumped phage particles…”. Since phage “lysates” were prepared“… by heating phage with reducing buffer…”, it is unlikely that phage particles formed clumps. Please address.

Lines 311-312: “…. however, the functional properties the bacterially expressed VLRB was further tested for immunohistochemistry (IHC) after another round of his-tag purification”. Not sure whether this statement is needed here. Please clarify/rephrase/remove.

Lines 331-332 and Figure 6: “Positive staining of the IgM cells by VLRB.IgM3P_4 was also detected using 4C4 (data not shown)”. Do the Authors want to include it in Figure 6?

Discussion

Lines 352-353: “Amplification of the high affinity binders in the panning rounds was observed in the case of both anti-lysozyme and anti-human IgM phage display screening”. Affinity of the binders was not measured. Do the Authors need to rephrase to describe relative/comparative affinity?

Line 362: Remove “data not shown”. It is the discussion section, not results.

Line 378: “… faithful immunization of the lampreys…” Not sure what “faithful immunization” is. Please clarify/rephrase.  

Materials and Methods

No statistics.

Line 440: Wrong place to reference Figure 1A. It is better to remove this Figure completely.

Line 466: Wrong place to reference Figure 1B.

Lines 481-484: To precipitate phage we use PEG, not blocking buffer. Please rephrase for clarity.

Line 538: “Panning was performed as shown schematically in Figure 2B”. There is no Figure 2B in the manuscript. Please clarify.

Line 541, 543:”… phage solution…”. Use the term phage “suspension”, not “solution”.

Line 545: “…spent library…” What is spent library? Please clarify.

Line 550: Tris-HCl?

Lines 565,584. Usually, filamentous phage stocks are stored in 50% glycerol, not 10%. Please clarify.

Lines 588, 592, 625: For consistency, use “g” instead of “rpm”.

Acknowledgements

Lines 683-685: “We thank Max Cooper, Emory University, for contributing to this project intellectually and for providing resources towards this work. Research conducted at Emory University was funded, in part, by NIH grant R01AI072435 (to Cooper)”. Should Dr. Cooper be a co-author?

References

Multiple references do not include journal names.

Author Response

This manuscript is a resubmission of an earlier submission. The following is a list of the peer review reports and author responses from that submission.

Round 1

Reviewer 1 Report

In this present study, the authors firstly tested the efficacy of the phage display antibody technology by isolating a previously screened anti-HEL antibody spiked in a very large antibody library and then obtained anti-HEL and anti-human IgM antibodies from the antibody libraries constructed from the lamprey larvae immunized with the target antigens using this novel technology.

1.     It deserves a better explanation on the rationale to define the OD reading above 0.1 as a positive reaction as shown in Figure 2. What is the OD reading of the positive clone isolated previously?

2.     The way authors presented the data is very tedious and redundant. For example, the data in Tables S3A and S3B should be combined and presented in one Table since they are duplicated experiments. It would be more comprehensive if the data was further combined with those in Figure 2 to avoid the redundancy.

3.     It is not clear why the PCR product of the same clone showed different size in Figures 2A (experiment 1) and C (experiment 2). Same concern for OD readings in Figure 2B and D.

4.     Please clarify the PCR size (filled triangle) in y axis in Figure 3. Also, the PFU of starting phages (109 in line 312 or 1010 in line 224) used in Figure 3 needs further verification.

5.     The reason for constructing the antibody library from the white blood cells of animal 1 should be further specified. Why not combined samples from 3 immunized animals?

6.     In general, at least 4-5 rounds of biopanning are carried out when applying phage display technology. It is wondered the reason why only 2 rounds of biopanning were performed for anti-HEL antibody library and 3 rounds for anti-human IgM.

7.     The clone numbers in each group as described in lines 370 to 375 needs further verification.

8.     DNA sequencing work nowadays is commercially available and feasible. Thus, it is concerned why only 7 out of 12 clones were sequenced and analyzed even though the clonal expansion has been observed as the authors claimed.

9.     Please clarify the PCR size (filled triangle) in y axis and explain why higher OD readings were observed in more diluted phage samples as shown in Figure 7.

10.  The data for the expression and purification of recombinant VLRB.IgM3P_4 should be presented in the manuscript.

11.  As stated, anti-HEL and anti-human IgM antibody libraries contains 9x104 and 1.6x108 primary recombinant clones, respectively. It is wondered any particular speculation for the big difference in library sizes.

12.  The phage display antibody technology has been well demonstrated as a novel and powerful method for isolating target-specific monoclonal antibodies (mAb). Thus, it is not so sure that the significance or purpose of testing the efficacy of the biopanning procedure to obtain the specific anti-HEL mAb clones diluted in a very huge antibody library as done by the authors.

13.  Collectively, it is suggested that the authors presented the data focusing on the generation of lamprey monoclonal antibodies using the phage display system as titled.

Minor:

1.     The statement of labeled PC1 and the clone ID in lines 653 and 654 was confusing!

2.     Please modify the sentence “Incubate was carried out at 37˚C for 1 hour.” in line 655.

3.     Please clarify the wording in the parenthesis (mention other ingradients) line 691.

4.     Please clarify the description (after 20 hours ………on ice for 30 minutes) in lines 694 to 696.

5.     Also, check again 70% output in line 698?

6.     The description for IHC experiment should be modified. For example, the reference for standard protocol should be included. Also, the source of control antibody should be indicated.

Reviewer 2 Report

The authors describe a wonderful phage display system to select VLRs of lampreys. The experiments are not presented in a clear way and are not using the normal terms used by the phage display/antibody engineering community. The pre-experiments with the naive library and the spiking experiments can be drastically shortened. It‘s a pity, that no anti-lysozyme antibodies can be selected from a naive VLRB phage display library. The real experiments are starting with the immune libraries (2.4). The description of some of the axises is idiosyncratic. In most ELISAs the negative controls are missing.

Suggestion:

- minimize pre-experiments > only some sentences + supplementary material

- immune library against lysozyme + IgM are core experiments (figure 8 is very good) please add the negative controls

- SDS-PAGE with lampribodies + WB against pIII to show the lampribody::pIII fusion protein is essential for QC of the phage particles

Detailed Comments:

Major Compulsory Revisions:

-In figure 1B: threenegative controlsaremissing: the binding of the VLRB.HEL.x phage on a control protein to see the background binding. Phage particles have the tendency to unspecific binding in higher concentrations. ScFv/Fab phage particles which have a non-folded antibody fragment on the surface will bind unspecifically by hydrophic interactions. I assume this will be the same for lampribodies, especially when a high percentage of clones have no full size lampribody insert.

Please use a different font, it‘s very hard to read and looks very fuzzy. I suggest also to give the binding curves for all four lampribodies in colors. I suggest solid lines for the binding on the antigen and dashed lines for binding on control protein.

-I do not understand the experiment with the competitor phage display library. You created a naive (= non-immune library) and showed that a lot of clones do not have a full size insert and you have no lampribodies against lysozyme. Please write it in a clear way and use the standard terms. This part can be summarized to 5-10 sentences and no figure is needed (figure 2 and 3 can be deleted and moved to supplementary. In figure 2B+D the binding on negative control is missing).

-I understood, that the „subtraction experiment“ is a spiking experiment. You spiked the naive library (what you called competitor library) with VLRB.HEL21 phage particles (you called these phage driver phage).  Please write it in a clear way and use the standard terms.

- figure 5: the x-axis is confusing. What is 1, 2, 3...? 100, 10-1, ... dilution? Please label it 10-1, 10-2(use the logarythmic scale)... to avoid confusion. The same for figure 1B.

- figure 6 and 7. Here, you used the code 1, 2... for two fold dilutions. Please use a normal logarythmic scale.

- figure 6: What is 10bLysA1P2_1 WB <> ...NB? Is WB the dilution on the antigen and NB the dilution on a control protein (the term NB is used in figure 7 for „no bait“?

- the dilution ELISA of all different lampribodies from the immune would be interesting (on lysozyme and control protein).

Minor Revisions:

- abstract „and observed clonal expansion of the high affinity binders“. What does this mean? The term „clonal expansion“ is normally used for proliferating antigen activated B-cell clones. When you mean it in the context of phage display, I suggest following text: „and observed an amplification of the high affinity binders in the panning rounds“

- Hyperphage has to be written with a capital „H“ because it is a proper name.

- Can you give some info about the antibody 4C4 against VRLB for the serum ELISA? What does it bind (a conserved part of VRLBs?)?

p { margin-bottom: 0.21cm; background: transparent none repeat scroll 0% 0%; }a:link { color: rgb(0, 0, 128); text-decoration: underline; }
